# The impact of the COVID-19 pandemic on oral anticoagulation adherence in patients with atrial fibrillation managed in primary care: Results from the PRIME Registry

Omer Atac[1,2]*, Volkan Aydin[3], Lars E. Peterson[4,5], Teresa M. Waters[6]

**1** Institute of Population and Social Research, Marmara University, Turkey, **2** Department of Public Health, International School of Medicine, Istanbul Medipol University, Turkey, **3** Department of Basic Health Sciences, Faculty of Dentistry, Marmara University, Turkey, **4** American Board of Family Medicine, Lexington, Kentucky, United States of America, **5** Department of Family and Community Medicine, College of Medicine, University of Kentucky, Lexington, Kentucky, United States of America, **6** School of Public Health, Augusta University, Augusta, Georgia, United States of America

* omer.atac@marmara.edu.tr

## Abstract

### Background

Oral anticoagulants (OACs) are critical for preventing stroke in patients with atrial fibrillation (AF), yet adherence and persistence remain suboptimal, especially in primary care settings. The COVID-19 pandemic introduced new barriers to care that may have further disrupted medication use. This study aimed to examine the impact of the COVID-19 pandemic on OAC adherence and persistence among patients with nonvalvular AF managed in primary care practices.

### Methods

In this retrospective cohort study, we used clinical records from the American Board of Family Medicine's (ABFM) PRIME Registry, for March 2017 through March 2023. We included 3,010 patients with documented AF and at least two OAC prescriptions prior to baseline. Persistence was assessed annually using a treatment anniversary method, and adherence was measured using the proportion of days covered (PDC), analyzed among persistent patients only. Interrupted time series and multivariable logistic regression models evaluated changes in quarterly persistence and factors associated with persistence and good adherence.

### Results

Persistence declined sharply during the pandemic from 49.9% in 2018–2019 to 8.2% in the pandemic year-1. Adherence also dropped, with >95% PDC decreasing from 46.0% pre-pandemic to 17.7% in the pandemic year-2. Interrupted time series

**Data availability statement:** Data for this study were accessed through the Stanford Center for Population Health Sciences (PHS) Data Core and are not publicly available. Access to the data can be granted to researchers who meet the data access requirements and obtain permission from the PHS Data Core. Requests for data access can be made through the PHS Data Portal at: https://med.stanford.edu/phs/data.html or by contacting phsdatacore@stanford.edu.

**Funding:** This study was supported by The Scientific and Technical Research Council of Türkiye (TUBITAK) 2219-International Postdoctoral Research Fellowship Program for Turkish Citizens. The funders had no role in study design, data collection and analysis, decision to publish, or preparation of the manuscript.

**Competing interests:** The authors have declared that no competing interests exist.

analysis showed a significant immediate drop in persistence. Compared to warfarin, NOAC use was associated with lower persistence in the pre-pandemic period, but higher persistence by the second year of the pandemic.

## Conclusions

The COVID-19 pandemic was associated with sustained declines in OAC adherence and persistence among AF patients in primary care. Targeted interventions including telemedicine, home-based care, and attention to high-risk subgroups are essential to maintain continuity of care and improve adherence during public health crises.

## Introduction

Atrial fibrillation (AF) is the most common sustained cardiac arrhythmia and is associated with a substantially increased risk of stroke and systemic embolism, contributing to significant morbidity and mortality [1]. Oral anticoagulants (OACs) are highly effective in reducing this risk, but poor adherence remains a critical barrier to achieving optimal outcomes [2,3]. Maintaining good medication adherence is particularly vital, as even small lapses are associated with increased mortality [4]. Non-vitamin K antagonist oral anticoagulants (NOACs), introduced in the past decade, were expected to improve adherence owing to their advantages over vitamin K antagonists (VKA) like warfarin, such as fewer dietary restrictions and no need for routine monitoring [3,5]. However, these improvements have not been widely observed in practice [6,7], and lower adherence to NOACs is associated with worse outcomes compared to warfarin [8].

Common barriers to OAC medication adherence include patient-related factors such as inadequate stroke awareness and concerns about bleeding, as well as challenges tied to practitioners and the healthcare system [9,10]. In primary care, adherence has historically been low, with fewer than one-third of patients persisting on OAC therapy at two years after initiation [11]. Studies indicate higher use of VKAs in primary care, a factor that may further limit adherence [12]. During the COVID-19 pandemic, these challenges were likely amplified by disruptions to healthcare delivery, including fear of infection, limited access to facilities, and medication shortages [13,14]. Nationwide US pharmacy claims, however, revealed a significant improvement in OAC adherence early in the pandemic, particularly with NOACs and among patients managed in cardiology practices [15]. This contrasting evidence highlights the need to better understand adherence patterns in primary care during the pandemic, given its diverse patient populations, higher reliance on VKAs, and inherent barriers to access [16]. In this study, we assessed the impact of the pandemic on OAC adherence in AF patients managed in primary care.

## Methods

### Data and study design

We conducted a retrospective cohort study using clinical records from the PRIME Registry of the American Board of Family Medicine (ABFM). PRIME is a certified

outpatient clinical quality data registry that collects electronic health record (EHR) data through automated feeds from participating primary care practices. Currently, PRIME includes data from over 3,000 primary care clinicians who are caring for patients in 1,250 practices across 50 states, encompassing more than 6.5 million patients and over 70 million visits. We accessed de-identified retrospective medical data on 05/04/2024. No identifying information was available to the authors at any point during or after the data collection process.

We defined March 15, 2020, as the onset of the COVID-19 pandemic, aligned with the nationwide emergency declaration of COVID-19 in the USA (Fig 1). To evaluate the potential impact of the pandemic on medication adherence, we defined a 6-year study period from March 15, 2017, to March 14, 2023 and divided this time into six distinct periods starting from March 15th of each year and ending on March 14th of the subsequent year: baseline year (Base): March 15, 2017 to March 14, 2018; pre-pandemic year-2 (Pre-2): March 15, 2018, to March 14, 2019; pre-pandemic year-1 (Pre-1): March 15, 2019, to March 14, 2020; pandemic year-1 (PY-1): March 15, 2020, to March 14, 2021; pandemic year-2 (PY-2): March 15, 2021, to March 14, 2022; and post-pandemic year-1 (Post-1). We primarily compared the pre-pandemic years with the first two pandemic years and the post-pandemic year. We included data from two years prior to the start of the pandemic and one year after to capture any trends that may have occurred during these periods.

## Study population

We defined a set of inclusion and exclusion criteria to ensure a comparable and continuously observed study population across the different time periods. First, we identified primary care practices that consistently provided a minimum level of service throughout the study period. We excluded practices that had no recorded patient visits for six months or more and those with fewer than 250 total visits in a single calendar year to include clinics actively providing continuous care throughout the study period. After these exclusions, our database included 1,281 primary care clinicians from 359 primary care practices across 40 states. Next, we categorized the oral anticoagulants of interest into NOACs: apixaban, dabigatran, edoxaban, and rivaroxaban; and a VKA: warfarin. We then established the inclusion criteria for patients. We selected patients who had documented history of AF in the EHR and had at least two prescriptions for the specified medications in the pre-study period (March 15, 2016, to March 14, 2017). This criterion ensured inclusion of patients who had been on treatment for a sustained period before baseline, allowing for more reliable assessment of longer-term medication usage. To assess medication adherence, we identified the first prescription for each patient within the baseline year (March 15,

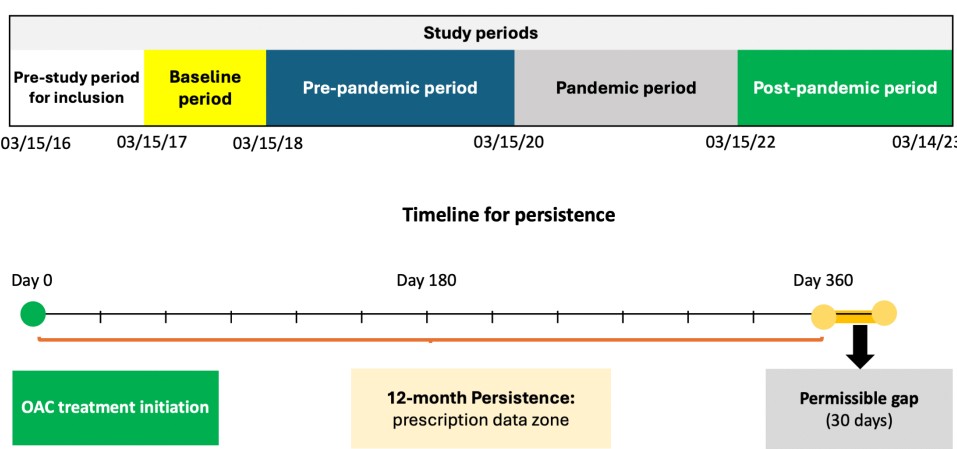

**Fig 1. Study design of the study and timeline for persistence calculations.**

2017, to March 14, 2018) and tracked prescription records for the following five years. We excluded patients with missing data on essential demographic and clinical variables, such as age, sex, or prescription dates and the proportion of missingness was low (<1%), so no imputation methods were necessary. After applying these criteria, the sample included 3,010 patients.

## Outcome measures and other study variables

Our outcome measures were persistence and adherence rates in each time period (Fig 1). Persistence and adherence were evaluated as distinct yet complementary dimensions of treatment continuity. In this study, we used the treatment anniversary method to measure persistence. A patient was considered persistent if they had an active prescription for the medication on the anniversary of the start date. We allowed a permissible gap of up to one month after the anniversary. Adherence was measured using the proportion of days covered (PDC) method for the year. Because adherence reflects day-to-day medication-taking behavior, PDC was computed only among patients who remained persistent during the corresponding observation year. This conditional approach distinguishes short-term medication-taking behavior from long-term therapy continuation. We restricted the adherence denominator to persistent patients to avoid underestimating adherence due to early discontinuers, who by definition have no coverage beyond discontinuation [17]. We stratified adherence rate levels (<80%, 80–84%, 85–89%, 90–94%, 95–100%) and differentiated among relatively high levels of adherence because even modestly imperfect adherence has been associated with poor outcomes in OAC use [4]. We categorized adherence rates as good for 80% and above, and as poor for below 80%. While this method provides a more precise estimate of medication-taking behavior, it may yield slightly higher adherence values and limit direct comparability with studies using unconditional or compliance-based definitions. In order to analyze trends in primary care-based persistence and adherence rates over time, we compared the rates across the five study time periods.

To account for patient characteristics, our independent variables included sex, race, ethnicity, rurality, and residence in historically marginalized areas. As documented in the EHR, patient race was categorized as white, black or African American, and other, while ethnicity was categorized as Hispanic or Latino versus non-Hispanic or Latino. We used the 2010 Rural-Urban Commuting Area (RUCA) codes, developed at the University of Washington, to classify patient zip codes as urban or rural [18]. These codes classify U.S. census tracts based on population density, urbanization, and commuting patterns. We assessed the residence in a historically marginalized area using the Social Deprivation Index (SDI) of the patient's county of residence [19]. The SDI is derived from socioeconomic indicators and reflects an individual's exposure to historical disinvestment in resources and marginalization. A patient's residence was classified as having a high degree of historical marginalization if the SDI for the county was higher than 50.

To characterize patients clinically, we calculated the Charlson Comorbidity Index (CCI) and the $CHA_2DS_2$–VASc score at treatment initiation for each patient based on diagnoses recorded prior to their first included prescription during the pre-study period (March 15, 2016 to March 14, 2017). CCI is a prognostic tool that predicts 10-year survival considering 17 categories of health conditions [20]. We categorized the CCI scores into three groups: mild risk (CCI scores of 1–2), moderate risk (CCI scores of 3–4), and severe risk (CCI scores of 5 or higher). The $CHA_2DS_2$-VASc Score calculates ischemic/stroke risk in patients with AF and is considered by physicians when deciding whether to initiate OACs. We categorized the $CHA_2DS_2$-VASc scores into two groups: low-intermediate (scores of 0–1 for males or 1–2 for females), and high (scores of 2 or higher for males and 3 or higher for females) [21].

## Statistical analyses

Descriptive and clinical characteristics of the patients were summarized as number and percentage. Persistence and adherence were reported as rates by study periods. We also developed an interrupted time series (ITS) model to assess the impact of the pandemic on quarterly persistence rates. We selected segmented linear regression for the ITS analysis due to its interpretability and suitability for our quarterly data and clear intervention point. This approach is widely used

in health services research, particularly for evaluating policy or system shocks with a moderate number of time points. We tested for autocorrelation in the residuals using the Durbin-Watson statistic and inspected autocorrelation and partial autocorrelation plots. No significant autocorrelation was detected (Durbin-Watson: 2.14). To provide clearer inference, we report 95% confidence intervals for model coefficients in the results. We defined the first quarter of the pandemic (March 15, 2020) as the intervention period and the rest of the study period (through 3/14/23) as post-intervention periods. Equation 1 outlines our empirical model.

$$Y_t = \beta_0 + \beta_1 t + \beta_2 D_{pandemic(t)} + \beta_3 T_{pandemic(t)} + \epsilon_t \tag{1}$$

In this model, $Y_t$ represents the quarterly persistence rate at time $t$. $\beta_0$ represents the initial level of quarterly persistence rates at the start of the study. $\beta_1$ captures the quarterly persistence rate trends during the pre-pandemic period. t is a time trend, representing each quarter during the study period. $\beta_2$ captures the immediate level change in persistence rates in the first quarter of the pandemic. $\beta_3$ captures persistence rate trends during the rest of the pandemic and the post pandemic period. $\epsilon_t$ is the error term. Additionally, to assess the association between independent variables and persistence and good adherence among patients classified as persistent in each time period, we developed a multivariable logistic regression model that included demographic and clinical variables.

Statistical analyses were performed using Stata 17.0 (StataCorp, College Station, TX, USA). Statistical significance was set at a two-sided p-value <0.05. This study was approved by the American Academy of Family Physicians (AAFP) Institutional Review Board (IRB Application #22–450; Approval Date: 08/30/2022; Amendment #1 Approval Date: 11/02/2022). The IRB waived the requirement for informed consent as the study involved secondary analysis of de-identified electronic health record data. No identifiable private information was accessed by the researchers. This waiver of consent complies with U.S. federal regulations for research involving secondary use of de-identified electronic health record data, as outlined under the Department of Health and Human Services (HHS) guidelines (45 CFR 46.104, category 4).

## Results

Our study sample included 3,010 individuals (Table 1). Almost two-thirds (66.3%) of our sample were between the ages of 65 and 84 years old, 55.8% were female, and 88.9% were non-Hispanic white. Most patients were urban residents (62.2%) and in the 0–50 (low) SDI category (52.2%). Warfarin was by far the most prescribed medication (93.1%). Based on CCI scores, 80.9% of the patients had severe comorbidities. Based on $CHA_2DS_2$-VASc scores, 91.0% of the patients were at high risk of ischemic stroke.

Persistence rates were highest during the Pre-2 period at 49.9%, and declined in the following periods: 28.3% in Pre-1, 8.2% in PY-1, 4.1% in PY-2, and 2.6% in Post-1 (Table 2). These rates reflect annual treatment anniversary persistence, defined as remaining on therapy with no > 30-day gap over a 12-month period. Adherence rates, calculated among persistent patients only, also declined overall. They started at 65.1% in Pre-2 and 64.1% in Pre-1, then dropped to 48.6% in PY-1. Although adherence appeared to slightly increase in PY-2 (50.8%) and Post-1 (57.0%), rates remained lower than pre-pandemic levels. However, these estimates are based on a very small subset of persistent patients (4.1% and 2.6% of the sample, respectively). Poor adherence was most common in PY-1 (51.4%) and PY-2 (49.2%), compared to 34.9% in Pre-2 and 35.9% in Pre-1. The proportion of patients with high adherence (PDC ≥ 95%) declined from 46.0% in Pre-1 to 26.7% in PY-1, 17.7% in PY-2, and 19.0% in Post-1.

Fig 2 displays quarterly persistence rates as modeled in the ITS analysis, which offers a higher temporal resolution compared to the annual treatment anniversary-based estimates. The vertical dashed red line denotes the start of the pandemic. We observed a significant immediate drop in quarterly persistence rates in Q2 2020 (coefficient: −24.99; 95% CI: −37.78 to −12.22; p = 0.001), relative to the expected rate based on the pre-pandemic trend. Although the post-intervention trend was slightly positive, it was not statistically significant (coefficient: 0.25; 95% CI: −2.11 to 2.61; p = 0.826).

**Table 1. Characteristics of the patients with Atrial Fibrillation in the PRIME Registry 2016 to 2022.**

| Age Group (n = 3,010) | n | % |
|---|---|---|
| ≤64 yrs | 409 | 13.6 |
| 65-84 yrs | 1,996 | 66.3 |
| ≥85 yrs | 605 | 20.1 |
| **Gender (n = 3,007)** | | |
| female | 1,330 | 44.2 |
| male | 1,677 | 55.8 |
| **Race (n = 2,814)** | | |
| white | 2,503 | 88.9 |
| non-Hispanic Black | 135 | 4.8 |
| Hispanic | 136 | 4.8 |
| other | 40 | 1.4 |
| **Social Deprivation Index (n = 3,002)** | | |
| 0-50 | 1,567 | 52.2 |
| 51-100 | 1,435 | 47.8 |
| **Rural-Urban Commuting Area (n = 3,002)** | | |
| urban | 1,867 | 62.2 |
| rural | 1,135 | 37.8 |
| **Medication (n = 3,010)** | | |
| warfarin | 2,802 | 93.1 |
| apixaban | 115 | 3.8 |
| rivaroxaban | 75 | 2.5 |
| dabigatran | 16 | 0.5 |
| edoxaban | 2 | 0.1 |
| **Charlson Comorbidity Index (n = 3,010)** | | |
| mild or moderate | 573 | 19.1% |
| severe | 2,432 | 80.9% |
| **$CHA_2DS_2$-VASc Risk Category (n = 3,002)** | | |
| low-intermediate | 270 | 9.0 |
| high | 2,732 | 91.0 |

**Table 2. Annual persistence and adherence rates of anticoagulation therapy by the study periods (n = 3,010).**

| | Pre-2 | Pre-1 | PY-1 | PY-2 | Post-1 |
|---|---|---|---|---|---|
| **Persistence, n (%)** | 1502 (49.9%) | 852 (28.3%) | 247 (8.2%) | 123 (4.1%) | 78 (2.6%) |
| Poor adherence, n (%) | 524 (34.9%) | 306 (35.9%) | 127 (51.4%) | 61 (49.2%) | 34 (43.0%) |
| Good adherence*, n (%) | 978 (65.1%) | 546 (64.1%) | 120 (48.6%) | 63 (50.8%) | 44 (57.0%) |
| *80-84, n (%)* | 78 (5.2%) | 55 (6.4%) | 18 (7.3%) | 13 (10.5%) | 7 (8.9%) |
| *85-89, n (%)* | 75 (5.0%) | 42 (4.9%) | 15 (6.1%) | 10 (8.1%) | 10 (12.7%) |
| *90-94, n (%)* | 79 (5.3%) | 58 (6.8%) | 21 (8.5%) | 18 (14.5%) | 13 (16.5%) |
| *95-100, n (%)* | 746 (49.6%) | 391 (46.0%) | 66 (26.7%) | 22 (17.7%) | 15 (19.0%) |

* Good adherence is defined as proportion of days covered ≥80%. Subcategories (in italics) indicate the distribution within the good adherence group.

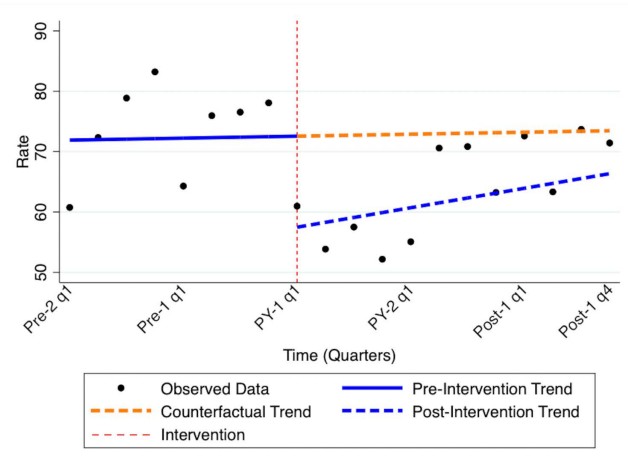

**Fig 2. Interrupted time series for persistence rates in each time period, quarterly.**

Among patients who were persistent, logistic regression models showed that those using NOACs had significantly lower odds of persistence in Pre-2 (OR: 0.35, 95% CI: 0.25–0.50) and Pre-1 (OR: 0.55, 95% CI: 0.30–0.98), but higher odds during PY-2 (OR: 12.50, 95% CI: 1.32–100.00) (Fig 3a). Patients who were classified as high risk based on their $CHA_2DS_2$–VASc score were more likely to be persistent in Pre-2 (OR: 1.64, 95% CI: 1.14–2.38). Patients residing in rural areas also had higher odds of persistence during PY-1 (OR: 2.44, 95% CI: 1.75–3.33) and PY-2 (OR: 2.56, 95% CI: 1.45–4.76).

Patients in the high SDI category were more likely to show good adherence in both Pre-2 (OR: 1.37; 95% CI: 1.09–1.72) and PY-1 (OR: 1.96; 95% CI: 1.11–3.45) (Fig 3b). Similarly, increasing rurality was associated with higher odds of good adherence during PY-1 (OR: 1.92; 95% CI: 1.08–3.45) and PY-2 (OR: 3.57; 95% CI: 1.37–9.09).

## Discussion

Our study sought to evaluate the impact of the COVID-19 pandemic on OAC adherence among atrial fibrillation patients across a sample of US primary care practices. We found that the pandemic was associated with a dramatic decline in medication persistence and adherence in this high-risk population, and we did not observe any significant rebound in these key outcomes in later pandemic time periods. Discontinuation and poor adherence were more pronounced among urban residents and those residing in historically marginalized areas.

Effective management of stroke prophylaxis in AF is threatened by underuse of OACs, a highly prevalent phenomenon [22,23]. While this has been mainly attributed to the suboptimal prescribing [24], OAC adherence has continued to decline over time [25] and has not improved with the introduction of NOACs [26]. We also observed dramatic declines in OAC persistence prior to the pandemic, declining from 49.9% in the Pre-2 period (2018–19) to 28.3% in the Pre-1 period (2019–20). This represents a much steeper drop than typically reported in the literature. A recent observational study of long-term adherence in incident AF patients reported a 12.5% annual decline in VKA users (>90% of our sample) [25]. Based on those data, our pre-pandemic decline (21.6% drop) is nearly double what might be expected. The sharper pre-pandemic decline in persistence compared with prior reports may reflect a combination of methodological and contextual factors. Because our study focused on established OAC users with prior prescriptions, the sample may have included a greater proportion of patients already approaching discontinuation. In addition, persistence was measured from primary care-EHR data, and prescriptions filled in specialty or external settings may not have been captured, which could understate

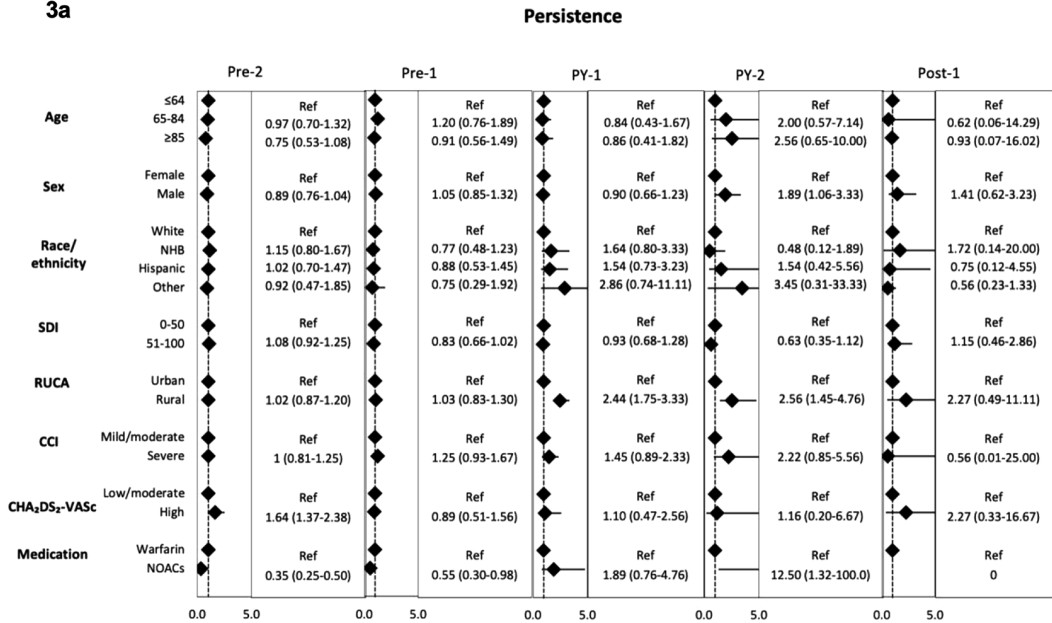

3a

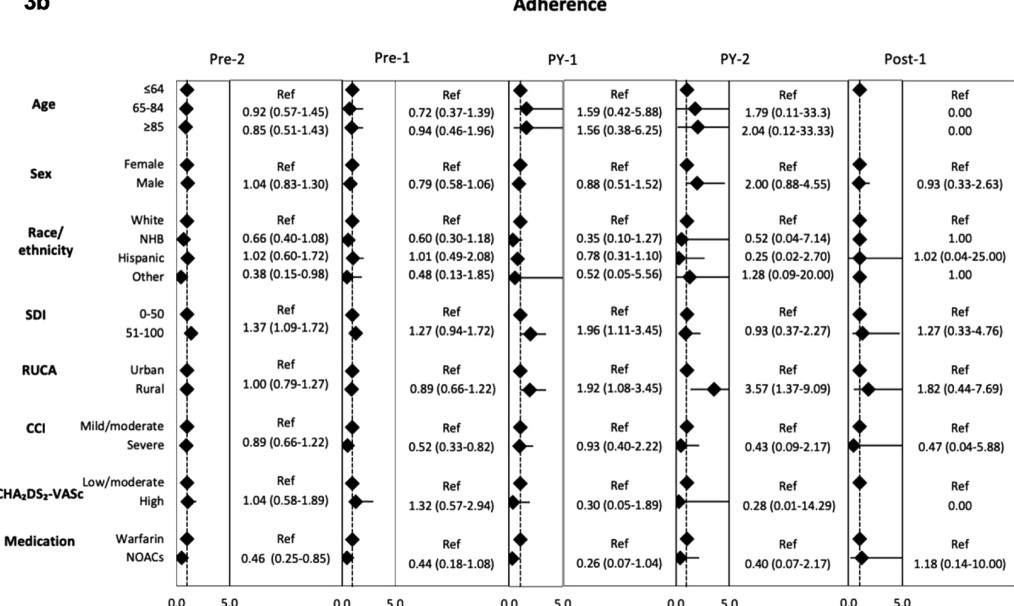

3b

**Fig 3. Logistic regression models indicating factors associated with persistence and good adherence among patients classified as persistent across study periods (n = 3,010).**

true persistence just before the pandemic. The sharper drop at the onset of the pandemic in our study likely reflects the major barriers patients faced in maintaining OAC therapy, including disruptions in healthcare access, fears of virus exposure, and logistical barriers during lockdowns and restrictions [14,27]. In addition, long-term OAC adherence trajectories showed rapid decline and discontinuation as responsible for near half of the nonadherent patients [28]. This aligns with the sharp and immediate drop in persistence that we observed at the onset of the pandemic in our ITS analysis, although

we did not detect significant trend changes during the later pandemic period. However, not all studies have reported similar findings. For instance, Hernandez et al. reported increased OAC possession during the first three months of the pandemic, particularly among NOAC users, based on claims data [15]. At first glance, these results appear inconsistent with ours, but the difference largely reflects variation in data source, care context, and study population: Claims data primarily capture dispensing and insurance transactions that measure medication possession rather than confirmed use, often encompassing prescriptions from cardiology, specialty, and hospital settings. In contrast, our primary-care EHR data capture longitudinal prescribing activity within community practices, offering a closer view of ongoing therapy management and care disruptions. Moreover, registry-based EHR cohorts such as ours typically include established, older users, whereas claims databases often represent mixed or newly treated populations. Thus, while claims data may show short-term refill surges or stockpiling early in the pandemic, practice-based EHR data reveal longer-term discontinuation and reduced treatment continuity. These complementary perspectives highlight how adherence and persistence trends can differ depending on data environment and clinical setting, reinforcing the need for contextual interpretation when comparing studies.

In our cohort of persistent patients, poor adherence rose from about one-third of our sample before the pandemic to about one-half during the pandemic and post-pandemic periods. This trend partly arises because adherence was evaluated only among patients who remained persistent under the anniversary-based definition; as overall persistence declined, the remaining cohort represented a progressively smaller and inherently more adherent group. Consequently, modest increases observed in some intermediate adherence categories (e.g., 80–94%) reflect the selective retention of highly adherent patients rather than a true improvement in adherence behavior. This persistence-conditioned pattern underscores that, despite apparently stable adherence within this subgroup, overall treatment continuity continued to deteriorate at the population level.

The conventional adherence threshold of ≥80% widely used in drug utilization research was applied in our study to ensure comparability with prior literature and meta-analytic standards [29]. However, this cutoff is largely empirical and may underestimate clinically meaningful nonadherence for oral anticoagulants. Even discontinuation periods as short as one week have been associated with mortality, regardless of the OAC used [30], underscoring the need for more stringent adherence benchmarks. Because of the short half-lives and narrow therapeutic windows of these agents, several recent studies have advocated adopting higher thresholds (≥90–95%) to better capture clinically optimal adherence [4,31]. In our sample, patients with ≥95% adherence fell from 50% to 27% in the first year of the pandemic, continuing to drop afterwards, suggesting that a substantial proportion failed to maintain the adherence level required for optimal anticoagulation benefit. This is particularly troubling in our study cohort, since 91% already had a high risk of developing ischemic stroke and 81% had severe risk of 10-year mortality.

Our results related to the predictors of adherence are consistent with the literature [32]. It is well-known that adherence to OACs is influenced by age, comorbidities, concomitant medications, and risk scores [31]. Recent studies have identified male sex, hospitalization, CCI, bleeding, and NOAC use as risk factors for OAC nonadherence whereas white race, high-risk in CHA2DS2VASc score and polypharmacy as protective factors [25,33]. In our study, we also observed the protective effect of high CHA2DS2VASc score against discontinuation and the increased risk of discontinuation and poor adherence associated with NOACs prior to the pandemic. However, these factors were not associated with significant changes in adherence after the start of the pandemic. Instead, we observed that patients from rural areas and those living in high SDI regions showed lower odds of discontinuation and poor adherence. One explanation for this result is that patients who faced challenges to medication adherence prior to the pandemic may have been better equipped to deal with the obstacles created by COVID-19. In fact, a UK survey investigating the medication adherence of patients with chronic conditions during the pandemic found that those in the highest socioeconomic group were more likely to report difficulty accessing medications and disruption to their medication habits [34]. Another US study investigating sociodemographic inequalities in delaying health care during the pandemic reported college-educated individuals as more likely to delay care

[35]. On the other hand, a review about the antihypertensive medication adherence among underrepresented racial and ethnic groups showed worsening of the preexisting barriers by the pandemic [36]. Our results could be driven by the large proportion of white patients in our sample, who may have experienced the pandemic differently (e.g., differential access to community support or different healthcare priorities). These differences merit further investigation, especially among the AF population.

Interestingly, we found that patients living in rural areas and those in high social deprivation areas demonstrated higher persistence and adherence during the pandemic. This finding contrasts with prior research showing that such populations often face greater structural barriers to care, including limited provider availability, transportation challenges, and lower health literacy [37,38]. One possible explanation is that rural and high-SDI patients may have relied on more stable, community-based practices that maintained continuity of care or used simplified medication access strategies (e.g., long-term refills or local pharmacy support) [39]. However, our data do not include information on delivery modality (in-person vs. telehealth), pharmacy type, or refill mechanisms, so this interpretation remains speculative. Moreover, the number of patients decreased in the later study periods, which may have influenced subgroup patterns, including wider confidence intervals and underrepresentation of some groups. Future studies should explore whether community resilience or practice-level adaptations contributed to these differences, and whether these trends hold in more complete datasets.

The findings of this study should be interpreted in light of its limitations. First, our retrospective design may be subject to unmeasured confounding. We addressed this by using ITS and logistic regression models to control for known confounders and confirmed that there was no multicollinearity of overlapping factors through variance inflation factor analysis which were lower than 5 in all subgroups. Second, while we used prescription data from EHRs, the lack of linkage to pharmacy claims may have led to misclassification of adherence or discontinuation status. Additionally, prescriptions filled outside the primary care setting such as cardiology or urgent care may not have been captured in our dataset. This may have resulted in underestimation of true persistence rates, particularly if patients continued medications through alternate providers during the pandemic. Given that our cohort size also declined in later time periods, this could reflect both actual attrition and reduced data completeness, potentially contributing to the steeper-than-expected drop in persistence. Additionally, adherence was calculated only among patients classified as persistent, an approach that enhances conceptual clarity by separating medication-taking behavior from treatment continuation but may also exclude early discontinuers, slightly overestimate adherence, and limit comparability with studies using unconditional definitions. Third, by including only patients with at least two prescriptions before the study period, we focused on established users, potentially limiting generalizability to new users. Fourth, the dominance of warfarin use (93%) in our sample may limit the applicability of our findings to settings where NOACs are more commonly prescribed. Fifth, the predominantly White sample (89%) may not reflect the experiences of more diverse populations. Some logistic regression results showed wide confidence intervals, particularly for subgroup comparisons involving NOAC use. These likely reflect small sample sizes in specific strata and should be interpreted with caution. Additionally, the lack of linkage to death or hospital data may have overestimated discontinuation, and fewer records in the post-pandemic period limit the robustness of findings beyond that point. Finally, as this study is based on U.S. primary care data, results may not generalize to other healthcare systems or countries. Future research should incorporate pharmacy claims data, capture primary non-adherence, and include more diverse and representative populations. Finally, while the sample includes a broad U.S. primary care population, generalizability to non-U.S. settings or different care contexts may be limited. Future research should include pharmacy data, explore primary non-adherence, and apply trajectory-based methods across broader healthcare settings to validate and extend our findings.

## Conclusions

We observed that the COVID-19 pandemic was associated with substantial declines in OAC persistence and adherence among AF patients in primary care, highlighting the urgent need for targeted interventions during public health crises. Strategies such as telemedicine, home-based care models, and patient-centered approaches to ensure safe and timely

healthcare access can help mitigate crisis-driven disruptions [40,41], particularly for patients living in urban areas and those from more socioeconomically advantaged areas. Understanding how the pandemic affected adherence may help primary care providers better anticipate and address patient needs during future healthcare disruptions. More broadly, our findings underscore that OAC adherence is a complex and dynamic behavior shaped by patient, provider, and system-level factors that interact across care settings and social contexts. We conclude that building resilient anticoagulation-care pathways therefore requires not only logistical preparedness but also recognition of these behavioral and contextual determinants to maintain long-term continuity of AF care and improve outcomes in high-risk populations.

## Author contributions

**Conceptualization:** Omer Atac, Volkan Aydin, Lars E. Peterson, Teresa M. Waters.

**Data curation:** Omer Atac.

**Formal analysis:** Omer Atac.

**Funding acquisition:** Omer Atac.

**Methodology:** Omer Atac, Volkan Aydin, Lars E. Peterson, Teresa M. Waters.

**Supervision:** Lars E. Peterson, Teresa M. Waters.

**Visualization:** Omer Atac.

**Writing – original draft:** Omer Atac, Volkan Aydin, Lars E. Peterson, Teresa M. Waters.

**Writing – review & editing:** Omer Atac, Volkan Aydin, Lars E. Peterson, Teresa M. Waters.

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
