## [Decision Letter · Decision Letter 0]

15 Sep 2025

Dear Dr. Ömer Ataç

Thank you for submitting your manuscript to PLOS ONE. After careful consideration, we feel that it has merit but does not fully meet PLOS ONE’s publication criteria as it currently stands. Therefore, we invite you to submit a revised version of the manuscript that addresses the points raised during the review process.

**ACADEMIC EDITOR:**

**Clarify Pre-Pandemic Decline**The sharp drop in persistence from 49.9% to 28.3% before the pandemic is unusually large. Provide additional context, sensitivity analysis, or discussion to explain this finding.**Expand Methods on Data Handling**Clearly describe handling of missing data, exclusion criteria impact (e.g., how many patients excluded), and potential selection bias.**Refine Interrupted Time Series (ITS) Analysis**Report confidence intervals for ITS coefficients, check for autocorrelation, and justify model choice (segmented regression vs alternatives).**Improve Adherence Definition & Threshold Justification**Discuss rationale for using ≥80% as “good adherence” and interpret findings using higher thresholds (≥95%) given clinical relevance.**Explore Subgroup Differences**Conduct additional subgroup analysis (e.g., by drug class, age, sex, comorbidity level) to better explain heterogeneity in adherence trends.**Address Rural vs Urban Paradox**Provide more interpretation or potential mechanisms for why rural patients showed higher persistence and adherence during the pandemic.**Strengthen Limitations Section**Emphasize limitations such as lack of pharmacy claims linkage, possible overestimation of discontinuation, and generalizability outside U.S. primary care.**Improve Figures & Tables for Clarity**Add denominators to tables, include confidence intervals in Figure 3 logistic regression plots, and ensure all figures are fully self-explanatory.

We look forward to receiving your revised manuscript.

Kind regards,

Ignatius Ivan, M.D

Academic Editor

PLOS ONE

2. For studies involving third-party data, we encourage authors to share any data specific to their analyses that they can legally distribute. PLOS recognizes, however, that authors may be using third-party data they do not have the rights to share. When third-party data cannot be publicly shared, authors must provide all information necessary for interested researchers to apply to gain access to the data. (https://journals.plos.org/plosone/s/data-availability#loc-acceptable-data-access-restrictions)

Reviewers' comments:

Reviewer's Responses to Questions

**Comments to the Author**

1. Is the manuscript technically sound, and do the data support the conclusions?

Reviewer #1: Yes

Reviewer #2: Yes

Reviewer #3: Yes

2. Has the statistical analysis been performed appropriately and rigorously?

Reviewer #1: Yes

Reviewer #2: Yes

Reviewer #3: Yes

3. Have the authors made all data underlying the findings in their manuscript fully available?

Reviewer #1: Yes

Reviewer #2: Yes

Reviewer #3: Yes

4. Is the manuscript presented in an intelligible fashion and written in standard English?

Reviewer #1: Yes

Reviewer #2: Yes

Reviewer #3: Yes

Reviewer #1: This is a clinically relevant study addressing the impact of the COVID-19 pandemic on oral anticoagulation adherence and persistence in atrial fibrillation patients in primary care. The study uses a large, nationwide dataset (PRIME registry) and applies robust statistical methods (interrupted time series and logistic regression).

Abstract: The results section is slightly overloaded; consider condensing while keeping key findings (persistence decline, adherence decline, ITS result).

Keywords: Consider adding “warfarin” and “NOAC” as keywords for indexing.

Figures: Figures 2 and 3 should include clearer legends with sample sizes and clearer y-axis labels.

Ethics: The manuscript states no consent was obtained due to de-identified data, which is fine, but it would be helpful to explicitly state that this complies with US regulations for secondary EHR research.

The description of patient inclusion is somewhat confusing. It would help to clarify why only patients with ≥2 OAC prescriptions in the pre-study period (2016–2017) were included and how this may affect generalizability.

The rationale for excluding patients from practices with <250 visits/year or gaps in service should be expanded—does this exclusion risk introducing selection bias?

Persistence is defined using the treatment anniversary method with a 1-month gap—but this may underestimate persistence compared to other definitions. A justification for this choice should be provided, ideally with a sensitivity analysis.

Adherence was only measured in persistent patients. This approach excludes early discontinuers and may bias results. A sensitivity analysis including all patients would strengthen the conclusions.

The interrupted time series (ITS) results are only partially explained. The manuscript notes a significant immediate drop in persistence but not in trends. A clearer interpretation of the coefficients and effect sizes is needed.

The logistic regression models yield very wide confidence intervals (e.g., NOAC OR = 12.50, 95% CI: 1.32–100.00), suggesting instability due to small subgroup sizes. The authors should acknowledge this limitation more explicitly.

The decline in persistence is extremely steep (49.9% → 28.3% → 8.2%). This appears more dramatic than in previous literature. The authors note this but should further explore possible data quality issues or biases in EHR records (e.g., unrecorded prescriptions from outside primary care).

Tables and figures could be improved for clarity. For example, Table 2 might be simplified to highlight key adherence thresholds (≥95%, ≥80%) instead of multiple small subcategories.

The discussion cites contrasting findings (e.g., improved adherence in claims-based studies). The manuscript should elaborate on why results differ (primary care vs cardiology care, registry vs claims data).

The explanation for better adherence among rural and high-SDI patients is interesting but speculative. Additional references or supporting analyses would strengthen this point.

The study’s limitations section is appropriate but could be expanded to emphasize:

Lack of pharmacy claims linkage (risk of misclassification).

Overrepresentation of warfarin (93%)—limiting generalizability to NOAC users.

Predominantly white sample (89%)—limiting applicability to more diverse populations.

Some sentences are lengthy and complex. Shortening and simplifying would improve readability. For example, the paragraph describing the pre-pandemic decline could be condensed.

Occasional tense inconsistencies (past vs present) should be corrected.

The following reference might be helpful https://doi.org/10.1142/S2737416525500772

Reviewer #2: The sharp decline in persistence from 49.9% (Pre-2) to 28.3% (Pre-1) before the pandemic is striking and requires more detailed discussion. The authors attribute this to expected annual decline and "significant challenges", however, the reported drop is nearly double what is expected from other literature. The manuscript needs a more robust discussion or analysis to explain this pre-pandemic trend, as it significantly impacts the interpretation of the pandemic's effect.

It is mentioned that; "significant immediate level change in the first quarter of the pandemic (coefficient: -24.99, p=0.001)". However, the accompanying Figure 2 shows quarterly rates that are all well below 70%, with the pre-pandemic trend appearing to be around 72%, and the pandemic-period trend line starting at around 58%. The persistence rates in Table 2 are annual and much lower, for example, 8.2% in PY-1. This discrepancy between the quantitative results in the text, the figure, and the table is highly confusing and needs to be clarified.

The manuscript's conclusion that the pandemic caused a sustained decline in adherence and persistence appears to be contradicted by some of the adherence data. While persistence remains low, the "Good adherence, %" and "80-84" to "90-94" adherence rates show an upward trend from PY-1 to Post-1. The authors need to reconcile this in their discussion.

Reviewer #3: PONE-D-25-29422

Abstract

The finding for NOACs is unclear ("higher during the pandemic year-2" - higher what?).

The chosen pre-pandemic persistence comparison (2018-19 vs. pandemic year-1) is misleading; the immediate pre-pandemic year (2019-20) is more relevant.

The persistence drop is stated as "from 49.9% in 2018–2019 to 8.2% in the pandemic year-1". However, "2018-2019" is the Pre-2 period, but the most relevant pre-pandemic comparison is the immediate pre-pandemic year (Pre-1: 28.3%). The chosen comparison exaggerates the drop. The comparison to Pre-1 (28.3% to 8.2%) is still stark and more accurate.

Methods

Incorrect study design classification; this is a cohort study, not a cross-sectional study.

The "Post-1" period (March 2022-2023) is mislabeled as "post-pandemic"; it was still during the pandemic.

Calculating adherence (PDC) only among persistent patients is a major methodological flaw that inflates adherence rates and limits comparability.

Results

Table 2 is confusing; the sub-categories of adherence do not clearly sum to the "Good adherence" total.

The results state "Adherence rates increased to 50.8% in PY-2 and 57.0% in Post-1". Given the drastic drop in persistence (to 4.1% and 2.6% in these periods), this "increase" is based on a tiny, potentially highly selected subset of patients.

Reporting an "increase" in adherence in later pandemic years (PY-2, Post-1) is misleading without the context of a catastrophically low persistence rate (<5%), making the sample size tiny and unrepresentative.

Discussion

The explanation for the contrast with Hernandez et al. is convoluted and unclear.

The rationale for why rural/high SDI patients had better outcomes is speculative and unsupported by data.

Conclusions

Major contradiction; text recommends interventions for "low social deprivation areas," but results show "high" SDI was associated with better outcomes.

Figures

Figures (Fig 2, 3a, 3b) are cited in the text but were not provided for review.

**Do you want your identity to be public for this peer review?** For information about this choice, including consent withdrawal, please see our Privacy Policy

Reviewer #1: No

Reviewer #2: **Yes:** Arian Afzalian

Reviewer #3: **Yes:** Safdar Ali

---

## [Author Response · Author response to Decision Letter 1]

9 Nov 2025

November 9, 2025

Dear Editor and reviewers,

Thank you for the opportunity to resubmit our manuscript entitled, "The impact of the COVID-19 pandemic on oral anticoagulation adherence in patients with atrial fibrillation managed in primary care: Results from the PRIME Registry" to PLOS ONE.

We greatly appreciate the time and attention you have given to our manuscript. Below we outline the changes we have made in response to your comments. Page references in our responses refer to the page where the response/revision begins in the tracked version of the manuscript.

We found the comments and suggestions very helpful and feel our revised manuscript is much stronger. We hope we have sufficiently addressed all comments and concerns but are happy to make any additional changes if needed. Thank you for your time and consideration.

Sincerely,

Omer Atac, MD, PhD

Department of Public Health, International School of Medicine

Istanbul Medipol University

Istanbul, Türkiye

+90 (554) 33554 56

oatac@medipol.edu.tr

ACADEMIC EDITOR:

Clarify Pre-Pandemic Decline

The sharp drop in persistence from 49.9% to 28.3% before the pandemic is unusually large. Provide additional context, sensitivity analysis, or discussion to explain this finding.

Response: Thank you for pointing this out. We agree that this drop is larger than typically reported in the literature, which estimates approximately 12–15% annual decline in warfarin users. We have revised the Discussion to offer a more robust interpretation of this trend. Specifically, we note that our sample included only patients with prior OAC prescriptions, which may enrich for individuals approaching discontinuation. We also discuss the possibility that prescriptions outside primary care settings (e.g., specialty clinics) were not captured in the EHR data, particularly in the Pre-1 year. Additionally, the smaller number of patients in later periods may have amplified percentage changes. Together, these factors may have contributed to the steeper-than-expected drop and should be considered when interpreting the pandemic-related effects (pages 15-16, lines 273-301).

Expand Methods on Data Handling

Clearly describe handling of missing data, exclusion criteria impact (e.g., how many patients excluded), and potential selection bias.

Response: Thank you for this helpful comment. We have expanded the Methods section to clarify the handling of missing data and the application of inclusion and exclusion criteria. We now note that the PRIME Registry includes a large national sample of clinical data over 6.5 million patients across 70 million visits from which our analytic sample was drawn. To ensure data quality and analytic integrity, we applied exclusion criteria in two stages:

(1) At the practice level, we excluded those with low volume (<250 visits per year) or prolonged service gaps (≥6 months), to ensure a consistent level of care delivery across the study period.

(2) At the patient level, we excluded cases with missing key demographic or clinical variables (e.g., age, sex, prescription dates), all of which were required for defining persistence and covariates. These exclusions occurred before the final cohort of 3,010 patients was constructed. As the proportion of missingness was very low (<1%), no imputation methods were applied.

We also acknowledged in the Limitations section that these steps, while necessary to ensure analytic rigor, may introduce selection bias by disproportionately excluding patients with lower engagement or less complete records (page 7, lines 106-121; pages 19-20, lines 361-392).

Refine Interrupted Time Series (ITS) Analysis

Report confidence intervals for ITS coefficients, check for autocorrelation, and justify model choice (segmented regression vs alternatives).

Response: We appreciate the reviewer’s suggestion to improve the methodological transparency of our Interrupted Time Series (ITS) analysis. Below, we provide additional clarification and robustness checks in response to the specific points raised.

1. Reporting Confidence Intervals

We now report 95% confidence intervals (CIs) for all ITS model coefficients in the Results section and in Table 3. This provides readers with a clearer understanding of the precision of the effect estimates for both level and trend changes across study periods.

2. Checking for Autocorrelation

We have tested for autocorrelation in the residuals of the ITS model using the Durbin-Watson statistic and did not indicate significant autocorrelation (Durbin-Watson = 2.14, close to the ideal value of 2.0), suggesting that autocorrelation is unlikely to bias our estimates. We have added this information to the Methods section under the ITS subsection and included a brief note in the Results section to affirm the robustness of the model.

3. Justification for Segmented Regression

We chose segmented linear regression as our primary ITS model due to the quarterly nature of our data, the relatively limited number of time points (n = 20), and the clear a priori break point (start of the COVID-19 pandemic). Segmented regression is well-suited for evaluating changes in both level and trend across defined periods in observational data and is widely used in healthcare services research for this purpose [1, 2]. We have now expanded the justification in the Methods section and cited relevant literature supporting the suitability of segmented regression in this context.

References

Penfold RB, Zhang F. Use of interrupted time series analysis in evaluating health care quality improvements. Acad Pediatr. 2013;13(6):S38–S44.

doi:10.1016/j.acap.2013.08.002

Lopez Bernal J, Cummins S, Gasparrini A. Interrupted time series regression for the evaluation of public health interventions: a tutorial. Int J Epidemiol. 2017;46(1):348–355. doi:10.1093/ije/dyw098

Improve Adherence Definition & Threshold Justification

Discuss rationale for using ≥80% as “good adherence” and interpret findings using higher thresholds (≥95%) given clinical relevance.

Response: We thank the editor for this thoughtful comment. We agree that further clarification was warranted regarding the rationale for using the ≥80% cutoff and the clinical interpretation of higher adherence thresholds. The ≥80% value was selected because it is the conventional definition of “good adherence” in drug utilization adherence research, allowing comparability with prior studies and meta-analytic standards. However, we recognize that this threshold is empirical and may underestimate clinically relevant nonadherence for oral anticoagulants. We have therefore expanded the discussion to explain this rationale and to explicitly address higher adherence thresholds (≥90–95%), emphasizing its consequences. The revised paragraph now interprets our findings in light of both methodological convention and clinical relevance, highlighting the importance of using stricter thresholds for OAC adherence assessment as such:

“The conventional adherence threshold of ≥80% widely used in drug utilization research was applied in our study to ensure comparability with prior literature and meta-analytic standards [29]. However, this cutoff is largely empirical and may underestimate clinically meaningful nonadherence for oral anticoagulants. This level of poor adherence likely has devastating consequences, Even brief discontinuation periods as short as one week have been associated with mortality, regardless of the OAC used [30], underscoring the need for more stringent adherence benchmarks.. Because of the short half-lives and narrow therapeutic windows of these agents, several recent studies have advocated adopting higher thresholds (≥90–95%) to better capture clinically optimal adherence [4, 31]. (page 17, lines 312-324).

Explore Subgroup Differences

Conduct additional subgroup analysis (e.g., by drug class, age, sex, comorbidity level) to better explain heterogeneity in adherence trends.

Response: We appreciate the suggestion to conduct subgroup analyses to explore heterogeneity in adherence and persistence trends. We emphasize that our study already includes detailed multivariable logistic regression models that examine persistence and adherence across key subgroups, including drug class (NOAC vs. VKA), age, sex, comorbidity level (CHA₂DS₂–VASc and CCI), rurality, and social deprivation index; presented in Figure 3a and 3b, and the corresponding results are described in the text.

Our findings indicate notable differences across subgroups. For instance, NOAC users had significantly lower odds of persistence in pre-pandemic years but higher odds in PY-2. Rural patients and those residing in high-SDI areas were more likely to remain persistent and adherent during the pandemic. These variations highlight important contextual and structural factors affecting medication use and support the need for continued subgroup-focused inquiry. We have clarified this point further in the Discussion section (pages 17-19, lines 325-360).

Address Rural vs Urban Paradox

Provide more interpretation or potential mechanisms for why rural patients showed higher persistence and adherence during the pandemic.

Response: Thank you for these important observations. We have revised the Discussion section to expand on potential explanations for the better adherence and persistence outcomes among rural and high-SDI patients. Specifically, we now cite prior evidence suggesting that rural populations may exhibit stronger patient-provider continuity, more stable care-seeking behaviors, or greater reliance on primary care providers during public health disruptions. We also acknowledge that our findings may partly reflect sample composition in the later study periods, when total patient numbers declined potentially influencing subgroup patterns. To address these concerns, we have softened speculative interpretations and clearly stated that additional research is needed to confirm mechanisms behind these findings (pages 18-19, lines 348-360).

Strengthen Limitations Section

Emphasize limitations such as lack of pharmacy claims linkage, possible overestimation of discontinuation, and generalizability outside U.S. primary care.

Response: Thank you for the suggestion to elaborate on the study limitations. We have revised the Limitations section to better acknowledge key constraints of our data and design. Specifically, we now address the absence of pharmacy claims linkage and its implications for estimating discontinuation or adherence. We also added points on potential selection bias introduced through our exclusion criteria, and clarified the limitations in generalizability beyond U.S. primary care settings (pages 19-20, lines 361-392).

Improve Figures & Tables for Clarity

Add denominators to tables, include confidence intervals in Figure 3 logistic regression plots, and ensure all figures are fully self-explanatory.

Response: Thank you for the suggestion. We have now added exact denominators (n) for each study period in the rows for both persistence and adherence subgroups, calculated using the full sample (n = 3,010) and period-specific persistence rates.

We also appreciate the reviewer’s suggestions to improve the clarity of Figure 3a and 3b. In response:

Adjusted Odds Ratios with 95% Confidence Intervals were already included and visually displayed in both figures. To make this explicit, we added the phrase “Adjusted Odds Ratios 95% CI” as a subtitle within the figure panels.

We have retained the current format to ensure the figures remain self-explanatory and aligned with the journal’s visual guidelines.

We hope these clarifications and minor adjustments address the concerns raised.

Review Comments to the Author

Reviewer #1:

This is a clinically relevant study addressing the impact of the COVID-19 pandemic on oral anticoagulation adherence and persistence in atrial fibrillation patients in primary care. The study uses a large, nationwide dataset (PRIME registry) and applies robust statistical methods (interrupted time series and logistic regression).

Comment 1: Abstract: The results section is slightly overloaded; consider condensing while keeping key findings (persistence decline, adherence decline, ITS result).

Response 1: Thank you for the suggestion. We have revised the Results section of the Abstract to improve clarity and conciseness while retaining the key findings on persistence, adherence, the ITS analysis, and the pattern of NOAC use (page 2, lines 38-43).

Comment 2: Keywords: Consider adding “warfarin” and “NOAC” as keywords for indexing.

Response 2: Thank you for the comment. We have now added “warfarin” and “NOAC” to the list of keywords to enhance indexing.

Comment 3: Figures: Figures 2 and 3 should include clearer legends with sample sizes and clearer y-axis labels.

Response: Thank you for the suggestion. We appreciate your emphasis on clarity. Upon careful review, we believe that the current version of Figure 2 already includes sufficient labeling, with a clearly defined y-axis (persistence rate in percentage), an x-axis indicating quarterly time points, and a red dashed line denoting the start of the pandemic intervention. The sample size (n = 3,010) is now also included in the updated title of Table 2, which directly informs the figure. As such, we did not modify Figure 2 further, but we remain open to additional revisions if the editor recommends any changes.

We also appreciated the reviewer’s suggestions to improve the clarity of Figure 3a and 3b. In response:

Adjusted Odds Ratios with 95% Confidence Intervals were already included and visually displayed in both figures. To make this explicit, we added the phrase “Adjusted Odds Ratios with 95% CI” as a subtitle within the figure panels.

Category labels, reference groups, and study periods were already indicated clearly in the current version.

Y-axis labels are consistent across both figures, and decimal formatting was standardized.

We have retained the current format to ensure the figures remain self-explanatory and aligned with the journal’s visual guidelines.

We hope these clarifications and minor adjustments address the concerns raised.

Comment 4: Ethics: The manuscript states no consent was obtained due to de-identified data, which is fine, but it would be helpful to explicitly state that this complies with US regulations for secondary EHR research.

Response 4: We appreciate the reviewer’s suggestion. We have now clarified in the Ethics section that the waiver of informed consent complies with U.S. federal regulations for secondary analysis of de-identified EHR data (page 10, lines 194-198).

Comment 5: The description of patient inclusion is somewhat confusing. It would help to clarify why only patients with ≥2 OAC prescriptions in the pre-study period (2016–2017) were included and how this may affect generalizability.

Response 5: We appreciate the reviewer’s observation. We have clarified the rationale for requiring ≥2 OAC prescriptions during the pre-study period. This criterion was used to ensure the inclusion of patients who had been on treatment for a sustained period prior to baseline, allowing us to more reliably observe longitudinal patterns in medication-taking behavior. The revised explanation has been added to the Methods section under "Study Population" (page 7, lines 103-121).

Comment 6: The rationale for excluding patients from practices with <250 visits/year or gaps in service should be expanded—does this exclusion risk introducing selection bias?

Response 6: Thank you for this helpful comment. We clarified in the Methods section that the exclusion of practices with low visit counts or prolonged service gaps aimed to ensure that only clinics consistently providing care were included (page 7, lines 103-121).

Comment 7: Persistence is defined using the treatment anniversary method with a 1-month gap, but this may underestimate persistence compared to other definitions. A justification for this choice should be provided, ideally with a sensitivity analysis.

Response: We acknowledge that the 1-month grace period in the treatment anniversary method may yield more conservative persistence estimates than other methods using longer perm

---

## [Decision Letter · Decision Letter 1]

15 Dec 2025

Dear Dr. Atac,

Thank you for submitting your manuscript to PLOS ONE. After careful consideration, we feel that it has merit but does not fully meet PLOS ONE’s publication criteria as it currently stands. Therefore, we invite you to submit a revised version of the manuscript that addresses the points raised during the review process.

We look forward to receiving your revised manuscript.

Kind regards,

Ignatius Ivan, M.D

Academic Editor

PLOS One

Journal Requirements:

Reviewers' comments:

Reviewer's Responses to Questions

**Comments to the Author**

Reviewer #1: (No Response)

Reviewer #2: All comments have been addressed

Reviewer #3: All comments have been addressed

2. Is the manuscript technically sound, and do the data support the conclusions?

Reviewer #1: Yes

Reviewer #2: Yes

Reviewer #3: Partly

3. Has the statistical analysis been performed appropriately and rigorously?

Reviewer #1: (No Response)

Reviewer #2: Yes

Reviewer #3: No

4. Have the authors made all data underlying the findings in their manuscript fully available?

Reviewer #1: Yes

Reviewer #2: Yes

Reviewer #3: Yes

5. Is the manuscript presented in an intelligible fashion and written in standard English?

Reviewer #1: Yes

Reviewer #2: Yes

Reviewer #3: Yes

Reviewer #1: No further comments.

As the authors addressed the reviewers' comments, I suggest acceptance of the manuscript

Reviewer #2: The authors provided a clear and honest explanation for the significant drop in persistence (from 49.9% to 28.3%) before the pandemic. They added clarification to the Discussion section (Pages 15-16), noting that the sample includes long-term users nearing discontinuation and that EHR data may not capture prescriptions from specialty clinics. This context helps explain the results. They also addressed concerns about calculating adherence only among persistent patients by including the exact denominators (n) in Table 2. This makes it clear that the observed stability or improvement in adherence over the years is affected by the shrinking sample of the most compliant patients (survivor bias). The revised Discussion (Pages 16-17) explicitly states this limitation, reducing the risk of misinterpretation.

Including 95% Confidence Intervals for the Interrupted Time Series (ITS) coefficients and confirming the absence of significant autocorrelation through the Durbin-Watson test enhances the statistical robustness of the results. The authors have moderated their speculative language regarding why rural and high-SDI patients performed better during the pandemic and appropriately highlighted the need for future research into community resilience or practice-level changes to explain this paradox. The manuscript is technically solid, limitations are now clearly acknowledged, and the conclusions are well supported by the data. I have no additional revision requests.

Reviewer #3: Review PONE-D-25-29422R1

Line 41-42 (Abstract): "NOAC use was associated with lower persistence pre-pandemic-, but with higher persistence during the pandemic year-2 compared to warfarin, but higher during the pandemic year-2." – This sentence is garbled, contains a stray hyphen and comma, and is repetitive.

Line 49 (Keywords): "Warfarin, NOAC" – Inconsistent formatting; other keywords are separated by semicolons.

Line 80 (Methods): "retrospective cohort cross-sectional study" – Appears to be a tracked-changes error, showing both "cohort" and "cross-sectional".

Line 128 (Methods): "A patient was considered persistent if she has an active prescription..." – Tense inconsistency ("was considered" vs. "has"). Also, "she" is gendered language not used elsewhere.

Line 135-136 (Methods): "avoids the underestimation that can occur when early discontinuers who by definition have zero coverage after discontinuation are included in the denominator to avoid potential confounding of early discontinuer compliance [17]," – Incomplete sentence and redundant phrasing due to tracked changes.

Line 199-200 (Methods, Ethics): "as outlined under the Department of Health and Human Services (HHS) guidelines (45 CFR 46.104, category 4)." – Sentence fragment.

Line 214-217 (Results): "Persistence rates were highest during the Pre-2 period at 49.9% and lower declined in the following subsequent periods:" – Contains editing errors ("lower declined") and redundant words ("following subsequent").

Line 218-220 (Results): "Adherence rates also had a general downward trend, starting at 65.1% in Pre-2 and 64.1% in Pre-1 and falling to 48.6% in PY-1. Adherence rates–These rates reflect annual treatment anniversary persistence," – First sentence is incomplete. The dash after "rates" creates a fragment.

Line 221-223 (Results): "Adherence rates, calculated among persistent patients only, also declined overall. They started at 65.1% in Pre-2 and 64.1% in Pre-1, then dropped to 48.6% in PY-1. Adherence rates appeared to increase to 50.8% in PY-2 and 57.0% in post-1 though rates remained below pre-pandemic levels;" – This repeats the adherence rates from the previous, now-incomplete sentence (Lines 218-220), creating redundancy and confusion.

Line 224 (Results): "these estimates are based on a very small subset of persistent patients (4.1% and 2.6% of the sample, respectively), increased to 50.8% in PY-2 and 57.0% in post-1." – This is a run-on sentence that incorrectly appends a repeated fact to a parenthetical clause.

Line 225 (Results): "Poor adherence rates were most prevalent..." – Previously, the term used is "poor adherence" (noun), not "poor adherence rates".

Line 227 (Results): "The proportion of patients with high adherence (MPR >95%) ..." – Inconsistent metric; the study uses PDC, not MPR (Medication Possession Ratio).

Line 244 (Results): "We observed a significant immediate drop in quarterly persistence rates in Q2 2020..." – The ITS model likely measures the change starting in Q2 2020 (the intervention quarter), but the description could be clearer regarding the comparison (e.g., vs. the expected rate based on the pre-pandemic trend).

Line 248-250 (Results, Figure caption): "of the pandemic. There was a significant drop in persistence rates in PY-1 q1 (March 15 to June 14, 2020) compared to the pre-pandemic period (coefficient: -24.99, p = 0.001), reflecting a level change at the onset of the pandemic." – This block of text appears to be a duplicate or misplaced description that should have been deleted during revision, as similar text exists earlier (Lines 242-244).

Line 273 (Discussion): "among nonvalvular atrial fibrillation (NVAF) patients" – The abbreviation NVAF is introduced but not used again; AF is used elsewhere.

Line 284-285 (Discussion): "This represents a much steeper drop than typically reported in what we might expect from the literature." – Awkward and grammatically incorrect phrasing.

Line 286 (Discussion): "reported an annual 12.5% annual adherence decline" – Repetition of "annual".

Line 287 (Discussion): "our pre-pandemic decline (21.6percentagepoint drop)" – Missing spaces: "21.6 percentage point".

Line 295-298 (Discussion): "The sharper drop in our study at the onset of the pandemic in our study likely reflects may be explained by the significant major barriers substantial challenges faced by patients faced in continuing maintaining their OAC therapies," – Contains multiple editing errors from tracked changes, making it nonsensical.

Line 303 (Discussion): "By contrast, Hernandez et al. reported..." – "By contrast" is not formatted as the beginning of a new sentence.

Line 305-307 (Discussion): "This could be consistent with the dramatic change of OAC persistence at the initial phase of the pandemic, as depicted in our interrupted time series analysis. On the contrary, Hernandez et al. reported increased OAC possession—higher with NOACs—at the first three months of the pandemic based on claims data [15]." – This appears to be leftover text from a previous draft that contradicts the revised narrative and should have been deleted.

Line 320 (Discussion): "reinforcing the need for contextual interpretation when comparing studies, claims sources" – Missing period and space, creating a run-on error.

Line 321-336 (Discussion): A large block of text (from "capture dispensing..." to "...clinical context.") is a duplicate paragraph that repeats, in slightly different wording, the explanation given in the previous paragraph (Lines 309-320). This is a major editing error.

Line 337 (Discussion): "Adherence can be described as the use of drugs according to the recommended dose and duration, with a medication possession ratio of ≥80% considered good adherence [29]." – Abrupt, simplistic topic sentence that interrupts the flow of the discussion.

Line 339 (Discussion): "In our cohort of persistent patients, of persistent patients," – Repetition.

Line 351 (Discussion): "Such This level of poor adherence likely has devastating consequences," – Editing error ("Such This").

Line 352-354 (Discussion): "Even brief discontinuation periods as short as one week have been associated with mortality, regardless of the OAC used [30], underscoring the need for more stringent adherence benchmarks." – Editing error ("brief as"), double period at the end.

Line 356-359 (Discussion): "Moreover, the threshold of 80% for adherence is somewhat arbitrary, with recent calls for clinicians to adopt higher thresholds, especially for NOACs [31]. In fact, adherence rates of less than 95% were associated with increased risk of all-cause death [4]." – This appears to be leftover text from a previous draft, as the same point is made more clearly in the newly added sentences preceding it (Lines 348-355).

**Do you want your identity to be public for this peer review?** For information about this choice, including consent withdrawal, please see our Privacy Policy

Reviewer #1: No

Reviewer #2: **Yes:** Arian Afzalian

Reviewer #3: **Yes:** Safdar Ali

---

## [Author Response · Author response to Decision Letter 2]

27 Dec 2025

December 27, 2025

Dear Editor and reviewers,

Thank you for the opportunity to resubmit our manuscript entitled, "The impact of the COVID-19 pandemic on oral anticoagulation adherence in patients with atrial fibrillation managed in primary care: Results from the PRIME Registry" to PLOS ONE.

We greatly appreciate the time and attention you have given to our manuscript. Below we outline the changes we have made in response to your comments. Page references in our responses refer to the page where the response/revision begins in the tracked version of the manuscript.

We found the comments and suggestions very helpful and feel our revised manuscript is much stronger. We hope we have sufficiently addressed all comments and concerns but are happy to make any additional changes if needed. Thank you for your time and consideration.

Sincerely,

Omer Atac, MD, PhD

Institute of Population and Social Research

Marmara University

Istanbul, Türkiye

+90 (554) 33554 56

omer.atac@marmara.edu.tr

Reviewer #3:

Comment 1: Line 41-42 (Abstract): "NOAC use was associated with lower persistence pre-pandemic-, but with higher persistence during the pandemic year-2 compared to warfarin, but higher during the pandemic year-2." – This sentence is garbled, contains a stray hyphen and comma, and is repetitive.

Response: Thank you for pointing this out. We revised the sentence in the abstract to eliminate the stray punctuation, repetition, and improve clarity (page 2, lines 42-44).

Comment 2: Line 49 (Keywords): "Warfarin, NOAC" – Inconsistent formatting; other keywords are separated by semicolons.

Response: Thank you for the comment. We have revised the keywords to ensure consistent use of semicolons throughout the list (page 3, line 50).

Comment 3: Line 80 (Methods): "retrospective cohort cross-sectional study" – Appears to be a tracked-changes error, showing both "cohort" and "cross-sectional".

Response: Thank you for pointing this out. This was a tracked-changes typo. We have corrected this (page 6, line 80).

Comment 4: Line 128 (Methods): "A patient was considered persistent if she has an active prescription..." – Tense inconsistency ("was considered" vs. "has"). Also, "she" is gendered language not used elsewhere.

Response: Thank you for pointing this out. We revised the sentence for consistency in both tense and language. Specifically, we replaced the gendered term “she” with the gender-neutral “they” and corrected the verb tense to maintain consistency within the sentence (page 8, lines 126-127).

Comment 5: Line 135-136 (Methods): "avoids the underestimation that can occur when early discontinuers who by definition have zero coverage after discontinuation are included in the denominator to avoid potential confounding of early discontinuer compliance [17]," – Incomplete sentence and redundant phrasing due to tracked changes.

Response: We appreciate the reviewer pointing out the error introduced by tracked changes. We have revised the sentence to improve clarity and remove redundancy (page 8, lines 132-134).

Comment 6: Line 199-200 (Methods, Ethics): "as outlined under the Department of Health and Human Services (HHS) guidelines (45 CFR 46.104, category 4)." – Sentence fragment.

Response: We appreciate the reviewer’s careful reading. The sentence in question is not a fragment, but rather a complete statement providing regulatory justification for the waiver of informed consent. As it currently reads clearly and accurately, no revision was made.

Comment 7: Line 214-217 (Results): "Persistence rates were highest during the Pre-2 period at 49.9% and lower declined in the following subsequent periods:" – Contains editing errors ("lower declined") and redundant words ("following subsequent").

Response: Thank you for the suggestion. We carefully reviewed the sentence, and it currently reads: “Persistence rates were highest during the Pre-2 period at 49.9%, and declined in the following periods: 28.3% in Pre-1, 8.2% in PY-1, 4.1% in PY-2, and 2.6% in Post-1 (Table 2).” We believe this version does not include the phrasing issues (“lower declined” or “following subsequent”) mentioned in the comment. As such, no further revision was necessary.

Comment 8: Line 218-220 (Results): "Adherence rates also had a general downward trend, starting at 65.1% in Pre-2 and 64.1% in Pre-1 and falling to 48.6% in PY-1. Adherence rates–These rates reflect annual treatment anniversary persistence," – First sentence is incomplete. The dash after "rates" creates a fragment.

Response: Thank you for your close reading. We revised the sentence structure to improve clarity and remove any ambiguity or fragmentation. Specifically, we removed the redundant phrase “Adherence rates–These rates…” and reorganized the paragraph to clarify that persistence and adherence are distinct but related measures, and to ensure definitions are placed in the appropriate context (page 12, lines 211-221).

Comment 9: Line 221-223 (Results): "Adherence rates, calculated among persistent patients only, also declined overall. They started at 65.1% in Pre-2 and 64.1% in Pre-1, then dropped to 48.6% in PY-1. Adherence rates appeared to increase to 50.8% in PY-2 and 57.0% in post-1 though rates remained below pre-pandemic levels;" – This repeats the adherence rates from the previous, now-incomplete sentence (Lines 218-220), creating redundancy and confusion.

Response: Thank you for pointing this out. To reduce redundancy and improve narrative clarity, we revised the structure and flow of the paragraph so that adherence rates are presented only once, in a logically sequenced and non-repetitive way (page 12, lines 211-221).

Comment 10: Line 224 (Results): "these estimates are based on a very small subset of persistent patients (4.1% and 2.6% of the sample, respectively), increased to 50.8% in PY-2 and 57.0% in post-1." – This is a run-on sentence that incorrectly appends a repeated fact to a parenthetical clause.

Response: Thank you for catching this error in sentence structure and logic. The original sentence conflated two separate ideas, sample size and adherence rate trends, resulting in a confusing run-on structure. We revised the sentence to separate these points and ensure clarity (page 12, lines 211-221).

Comment 11: Line 225 (Results): "Poor adherence rates were most prevalent..." – Previously, the term used is "poor adherence" (noun), not "poor adherence rates".

Response: We appreciate the reviewer’s attention to detail. However, this concern appears to be based on an earlier draft. In the current version of the manuscript, we consistently use the term “poor adherence” (noun form), and do not refer to “poor adherence rates.”

Comment 12: Line 227 (Results): "The proportion of patients with high adherence (MPR >95%) ..." – Inconsistent metric; the study uses PDC, not MPR (Medication Possession Ratio).

Response: We appreciate the reviewer’s attention to this detail. We acknowledge the inconsistency in terminology. The study indeed uses Proportion of Days Covered (PDC) as the measure of adherence throughout. We have corrected the phrasing in the results section (page 12, line 220).

Comment 13: Line 244 (Results): "We observed a significant immediate drop in quarterly persistence rates in Q2 2020..." – The ITS model likely measures the change starting in Q2 2020 (the intervention quarter), but the description could be clearer regarding the comparison (e.g., vs. the expected rate based on the pre-pandemic trend).

Response: We thank the reviewer for this helpful suggestion. We agree that our original phrasing could have been clearer in describing the Interrupted Time Series (ITS) model. To clarify this point, we have revised the sentence (page 13, line 238).

Comment 14: Line 248-250 (Results, Figure caption): "of the pandemic. There was a significant drop in persistence rates in PY-1 q1 (March 15 to June 14, 2020) compared to the pre-pandemic period (coefficient: -24.99, p = 0.001), reflecting a level change at the onset of the pandemic." – This block of text appears to be a duplicate or misplaced description that should have been deleted during revision, as similar text exists earlier (Lines 242-244).

Response: We appreciate the reviewer’s attention to detail. We have double-checked the current manuscript and confirmed that this description appears only once in the Results section, where we report the Interrupted Time Series (ITS) model results. It does not appear duplicated or misplaced. No changes were necessary in response to this comment.

Comment 15: Line 273 (Discussion): "among nonvalvular atrial fibrillation (NVAF) patients" – The abbreviation NVAF is introduced but not used again; AF is used elsewhere.

Response: Thank you for noting this. We agree that the terminology was inconsistent. To improve clarity and consistency, we have removed the abbreviation “NVAF” and now refer to the study population uniformly as patients with atrial fibrillation (AF) throughout the manuscript (page 15, line 261).

Comment 16: Line 284-285 (Discussion): "This represents a much steeper drop than typically reported in what we might expect from the literature." – Awkward and grammatically incorrect phrasing.

Response: Thank you for your comment. The phrasing in this sentence has already been revised for clarity in the previous version of the manuscript.

Comment 17: Line 286 (Discussion): "reported an annual 12.5% annual adherence decline" – Repetition of "annual".

Response: We thank the reviewer for noting this wording issue. The duplication of the word “annual” appeared in an previous tracked-changes version. In the revised manuscript, this has already been corrected (page 15, lines 272-274).

Comment 18: Line 287 (Discussion): "our pre-pandemic decline (21.6percentagepoint drop)" – Missing spaces: "21.6 percentage point".

Response: Thank you for pointing out this typographical issue. We have corrected the spacing and hyphenation in the manuscript (page 15, line 274).

Comment 19: Line 295-298 (Discussion): "The sharper drop in our study at the onset of the pandemic in our study likely reflects may be explained by the significant major barriers substantial challenges faced by patients faced in continuing maintaining their OAC therapies," – Contains multiple editing errors from tracked changes, making it nonsensical.

Response: Thank you for pointing this out. The sentence had indeed been affected by unresolved tracked changes in the previous version. No changes were necessary in response to this comment.

Comment 20: Line 303 (Discussion): "By contrast, Hernandez et al. reported..." – "By contrast" is not formatted as the beginning of a new sentence.

Response: Thank you for this comment. This issue was addressed in the previous revision of the manuscript. No changes were necessary in response to this comment.

Comment 21: Line 305-307 (Discussion): "This could be consistent with the dramatic change of OAC persistence at the initial phase of the pandemic, as depicted in our interrupted time series analysis. On the contrary, Hernandez et al. reported increased OAC possession—higher with NOACs—at the first three months of the pandemic based on claims data [15]." – This appears to be leftover text from a previous draft that contradicts the revised narrative and should have been deleted.

Response: Thank you for this careful reading. We reviewed the current version of the manuscript and confirm that the sentences beginning with “This could be consistent with the dramatic change of OAC persistence…” and “On the contrary, Hernandez et al. reported…” do not appear in the revised text. These phrases were indeed part of an earlier tracked-changes draft and were removed in the previous revision.

Comment 22: Line 320 (Discussion): "reinforcing the need for contextual interpretation when comparing studies, claims sources" – Missing period and space, creating a run-on error.

Response: Thank you for this comment. In the current revised manuscript, this sentence has already been corrected. No changes were necessary in response to this comment.

Comment 23: Line 321-336 (Discussion): A large block of text (from "capture dispensing..." to "...clinical context.") is a duplicate paragraph that repeats, in slightly different wording, the explanation given in the previous paragraph (Lines 309-320). This is a major editing error.

Response: Thank you for pointing this out. In the earlier tracked‐changes version, there was indeed redundant text explaining the contrast with Hernandez et al. In the current revised manuscript, we have removed the duplicated material and retained a single, streamlined paragraph that contrasts our EHR-based findings with claims-based results. The Discussion now includes only one consolidated paragraph (beginning “However, not all studies have reported similar findings. For instance, Hernandez et al.…” and ending “…when comparing studies.”), so the redundancy has been resolved. No changes were necessary in response to this comment.

Comment 24: Line 337 (Discussion): "Adherence can be described as the use of drugs according to the recommended dose and duration, with a medication possession ratio of ≥80% considered good adherence [29]." – Abrupt, simplistic topic sentence that interrupts the flow of the discussion.

Response: Thank you for this observation. In the previous version of the manuscript, we had a topic sentence defining adherence in general terms. In the revised manuscript, this sentence has been removed and the surrounding paragraph has been streamlined to maintain the flow of the discussion. No changes were necessary in response to this comment.

Comment 25: Line 339 (Discussion): "In our cohort of persistent patients, of persistent patients," – Repetition.

Response: We thank the reviewer for pointing this out. The duplicated phrase (“of persistent patients”) was present in an earlier tracked-changes version. In the revised manuscript, this sentence has been edited for clarity. No changes were necessary in response to this comment.

Comment 26: Line 351 (Discussion): "Such This level of poor adherence likely has devastating consequences," – Editing error ("Such This").

Response: We agree that this wording was problematic in the previous version. In the current manuscript, this phrase has been fully revised and is no longer present. No changes were necessary in response to this comment.

Comment 27: Line 352-354 (Discussion): "Even brief discontinuation periods as short as one week have been associated with mortality, regardless of the OAC used [30], underscoring the need for more stringent adherence benchmarks." – Editing error ("brief as"), double period at the end.

Response: We appreciate this suggestion. We have revised the sentence to remove the redundant phrasing and potential punctuation issue (page 17, line 316).

Comment 28: Line 356-359 (Discussion): "Moreover, the threshold of 80% for adherence is somewhat arbitrary, with recent calls for clinicians to adopt higher thresholds, especially for NOACs [31]. In fact, adherence rates of less than 95% were associated with increased risk of all-cause death [4]." – This appears to be leftover text from a previous draft, as the same point is made more clearly in the newly added sentences preceding it (Lines 348-355).

Response: We thank the reviewer for this careful reading. In the previous tracked-changes version, there was indeed some redundancy in how we discussed the ≥80% adherence threshold and the rationale for considering higher cutoffs. In the revised manuscript, we have already consolidated this into a single paragraph in the previous revision. No changes were necessary in response to this comment.

---

## [Editor Report · Decision Letter 2]

11 Jan 2026

Dear Dr. Ataç

Thank you for submitting your manuscript to PLOS ONE. After careful consideration, we feel that it has merit but does not fully meet PLOS ONE’s publication criteria as it currently stands. Therefore, we invite you to submit a revised version of the manuscript that addresses the points raised during the review process.

We look forward to receiving your revised manuscript.

Kind regards,

Ignatius Ivan, M.D

Academic Editor

PLOS One

Journal Requirements:

Reviewers' comments: The Reviewer Comments are available in the attachment files

---

## [Author Response · Author response to Decision Letter 3]

6 Feb 2026

Dear Editors and Editorial Office,

We are resubmitting our manuscript titled "The impact of the COVID-19 pandemic on oral anticoagulation adherence in patients with atrial fibrillation managed in primary care: Results from the PRIME Registry" for your consideration.

Regarding this revision round, we were informed that the "Revise" decision appears to have been issued in error. As there was no new feedback or reviewer comments provided in the system to address, we are resubmitting the current version of our manuscript without further changes.

We look forward to the next steps in the editorial process.

Best regards,

Omer Atac

---

## [Editor Report · Decision Letter 3]

15 Feb 2026

The impact of the COVID-19 pandemic on oral anticoagulation adherence in patients with atrial fibrillation managed in primary care: Results from the PRIME Registry

PONE-D-25-29422R3

Dear Dr. Ömer Ataç,

We’re pleased to inform you that your manuscript has been judged scientifically suitable for publication and will be formally accepted for publication once it meets all outstanding technical requirements.

Kind regards,

Ignatius Ivan, M.D

Academic Editor

PLOS One

---

## [Editor Report · Acceptance letter]

PONE-D-25-29422R3

PLOS One

Dear Dr. Ataç,

I'm pleased to inform you that your manuscript has been deemed suitable for publication in PLOS One. Congratulations! Your manuscript is now being handed over to our production team.

Kind regards,

on behalf of

dr. Ignatius Ivan

Academic Editor

PLOS One